# A Novel Cytological Model of B-Cell/Macrophage Biphenotypic Cell Hodgkin Lymphoma in *Ganp*-Transgenic Mice

**DOI:** 10.3390/cancers12010204

**Published:** 2020-01-14

**Authors:** Yasuhiro Sakai, Andri Rezano, Seiji Okada, Takahiro Ohtsuki, Yoshiaki Kawashima, Tetsuya Tsukamoto, Motoshi Suzuki, Michinori Kohara, Motohiro Takeya, Nobuo Sakaguchi, Kazuhiko Kuwahara

**Affiliations:** 1Department of Diagnostic Pathology, Fujita Health University School of Medicine, Aichi 470-1192, Japan; ya-sakai@fujita-hu.ac.jp (Y.S.); ykawa@fujita-hu.ac.jp (Y.K.); ttsukamt@gmail.com (T.T.); 2Department of Biomedical Sciences, Division of Cell Biology, Faculty of Medicine, Universitas Padjadjaran, West Java 45363, Indonesia; andri.rezano@unpad.ac.id; 3Division of Hematopoiesis, Joint Research Center for Retroviral Infection, Kumamoto University, Kumamoto 860-0811, Japan; okadas@kumamoto-u.ac.jp; 4Department of Microbiology and Cell Biology, Tokyo Metropolitan Institute of Medical Science, Tokyo 156-8506, Japan; ohtsuki.tk@gmail.com (T.O.); kohara-mc@igakuken.or.jp (M.K.); 5Department of Molecular Oncology, Fujita Health University School of Medicine, Aichi 470-1192, Japan; motosuzu@fujita-hu.ac.jp; 6Department of Cell Pathology, Graduate School of Medical Sciences, Kumamoto University, Kumamoto 860-8556, Japan; takeya@kumamoto-hsu.ac.jp; 7Department of Immunology, Graduate School of Medical Sciences, Kumamoto University, Kumamoto 860-8556, Japan; nobuosakaguchi@gmail.com

**Keywords:** B-cell/macrophage biphenotypic cell, germinal center, germinal center-associated nuclear protein, Hodgkin lymphoma

## Abstract

Hodgkin lymphoma (HL) is one of the most difficult neoplasms in terms of cytopathological research owing to the lack of established cytological murine models. Although HL is believed to be of lymphoid germinal center B-cell origin, HL cells exhibit unique biphenotypic characteristics of B cells and macrophages. B-cell/macrophage biphenotypic cells have also been identified in the spleen of Lyn-deficient mice. Moreover, Lyn-targeting germinal center-associated nuclear protein (GANP)-transgenic mice (*Ig-ganp*^Tg^ mice) spontaneously develop a lymphoid tumor. We aimed to investigate whether the lymphoid tumor developed in *Ig-ganp*^Tg^ mice exhibit biphenotypic characteristics of B cells/macrophages that correspond to human HL. Here, we demonstrated GANP overexpression in human HL cells and found that it may regulate transdifferentiation between B cells and macrophages. We also demonstrated that tumors were comparable with B-cell/macrophage biphenotypic Hodgkinoid lymphomas. The tumor cells expressed macrophage-related F4/80, CD68, and CD204 as well as cytoplasmic B220 and µ-/κ-chains; in addition, these cells exhibited phagocytic activity. These cells also expressed transcripts of *CD30*; *c-fms*; and the cytokines monocyte chemoattractant protein (MCP)-1, MCP-5, RANTES, tumor necrosis factor-α and thrombopoietin associated with macrophages as well as granulocyte/macrophage colony-stimulating factor, interleukin (IL)-4, IL-10, IL-12, and IL-13. *Ig*-*ganp*^Tg^ mice represent a novel cytological model for the study of cytopathological etiology and oncogenesis of HL.

## 1. Introduction

Recent years have witnessed spectacular advances in the fields of molecular cytogenetics and cytopathology. These fields serve as a bridge between diagnostic cytology and molecular studies. Advanced cytological techniques such as immunocytochemistry, flow cytometry, liquid biopsy, and next-generation sequencing have helped revolutionize research on the etiopathogenesis and pathological diagnosis of neoplasms.

Cytological molecular analyses of Hodgkin lymphoma (HL) are typically challenging owing to the lack of established cytological murine models. HL is a lymphoid neoplasm characterized by the development of neoplastic giant cells called Hodgkin and Reed–Sternberg (HRS) cells; these cells originate from B cells located at the germinal center of peripheral lymphoid organs and may or may not be associated with Epstein–Barr virus (EBV) infection. HL cells exhibit rearrangement of immunoglobulin (*Ig*) genes and mutations of variable (*V*)-region genes; this indicates that their oncogenic transformation occurs at the germinal centers during somatic hypermutation of *IgV-region*. More interestingly, HL cells exhibit altered characteristics such as partial loss of B-cell lineage markers including surface IgM (sIgM), CD20, and CD79a; in addition, some HL cells express biphenotypic characteristics of B cells and macrophages [1].

The cytological characteristics of B-cell/macrophage biphenotypic cells are quite remarkable. Studies have shown that germinal center B cells can be reprogrammed for differentiation into macrophages by forced expression of C/EBP, which represses the expression of B cell-specific genes and activates macrophage-specific transcription factors [2]. Thus, germinal center B cells may retain hematopoietic plasticity for circumjacent immune response. These cells have been identified in human pathological conditions including HL, acquired immunodeficiency syndrome, and Sjogren syndrome [3]. Moreover, in some studies, B-cell/macrophage biphenotypic cells isolated from B-lymphoid leukemic patients were found to exhibit B-lymphoid and myeloid traits [4]. Thus, B-cell/macrophage biphenotypic cells are relevant to certain lymphoid neoplasms originating from B cells, particularly from germinal center B cells.

In src-type tyrosine kinase Lyn-deficient (*lyn^−/−^*) mice, B-cell/macrophage biphenotypic cells frequently appear in the spleen, germinal centers disappear in the peripheral lymphoid organs, and antibody affinity decreases [5]. Lyn is involved in signal transduction of intracytoplasmic molecules (such as sIgM and CD40) expressed on B cells in the peripheral lymphoid organs [6]. Germinal center-associated nuclear protein (GANP) is one of the targets of Lyn-mediated signaling in germinal center B cells for high antibody affinity maturation [7,8,9]. Abnormal overexpression of GANP has been reported in various human hematopoietic and lymphoid neoplasms including HRS cells in HL [10]; this suggests that Lyn-target GANP plays a regulatory role in the transdifferentiation or reprograming of B cells to macrophage-like cells. In addition, GANP plays a role in the development of B-cell/macrophage biphenotypic Hodgkinoid lymphoma, which corresponds to human HL.

Here, we revealed that GANP is immunocytochemically expressed on human HRS cells. We also revealed that GANP may regulate cell transdifferentiation of B cells to macrophages using *ganp*-transgenic mice under the control of Ig promoter and intron enhancer Eµ (*Ig*-*ganp*^Tg^ mice). Subsequently, we showed that *Ig*-*ganp*^Tg^ mice spontaneously develop Hodgkinoid lymphomas, which have similar properties as B-cell/macrophage biphenotypic cells such as phagocytic activity and macrophage-derived cytokine secretion. The novel cytological model of HL is based on the biphenotypic characteristics of B cells/macrophages; thus, it is very useful in studying the cytopathological etiology and oncogenesis of HL.

## 2. Results

### 2.1. Expression of GANP on Human Hodgkin Lymphoma Cells

We performed immunocytochemistry using the human HL cell lines HD70 and L428. Both cell lines showed strong expressions of GANP as well as CD15 and CD30, the diagnostic markers of HL (Figure 1). This result suggests that GANP is overexpressed on HRS cells and that it may be associated with Hodgkin lymphomagenesis.

### 2.2. Lyn-Independent Regulation of Differentiation to B-Cell/Macrophage Biphenotypic Cells in the Spleen by GANP

B-cell/macrophage biphenotypic cells have been previously identified in the spleen of *lyn^−^*^/*−*^ mice [5]. Our study newly indicates that HL cells express Lyn-target GANP. Therefore, we performed flow cytometry of surface antigens expressed by B cells and macrophages isolated from the bone marrow and spleen of *lyn*^−/−^ mice.

First, we compared bone marrow-derived B cells among 14-week-old *lyn*^−/−^ control, *lyn^−^*^/*−*^, and *Ig*-*ganp*^Tg^*lyn^−^*^/*−*^ mice. B cells with early differentiation such as B220^low^CD43^high^ (pro-B), B220^high^CD43^low^ (pre-B), and B220^low^CD43^low^ (immature B) cells as well as sIgM^+^surface IgD (sIgD)*^−^* immature B cells showed almost similar frequencies among these mutant and control mice; however, the frequency of sIgM^+^sIgD^+^ mature B cells was markedly lower in the bone marrow of *lyn^−^*^/*−*^ mice (Figure 2A) [6]. Additionally, in *Ig*-*ganp*^Tg^*lyn^−^*^/*−*^ mice, the frequency of sIgM^+^sIgD^+^ mature B cells did not recover. These results indicate that GANP may not influence early B-cell differentiation but may contribute to late-stage B-cell development in a Lyn-dependent manner.

Next, we analyzed the frequency of biphenotypic cells that express both B cell-specific marker Ig and macrophage-specific marker CD11b in these mice. A marked increase in sIgM^−^CD11b^+^ cells was observed in the spleen of *lyn^−^*^/*−*^ mice compared with that in the spleen of *lyn*^+/*−*^ mice (Figure 2B). More interestingly, cytoplasmic IgM (cIgM)^+^ cells were scarcely observed in the CD11b^+^ cell population in the spleen of eight-week-old *lyn^−^*^/*−*^ mice (Figure 2C); in contrast, approximately one-third of CD11b^+^ cells in the spleen of 14-week-old *lyn^−^*^/*−*^ mice were cIgM^+^ (Figure 2D). This indicates the appearance of cIgM^+^/CD11b^+^ B-cell/macrophage biphenotypic cells in *lyn^−^*^/*−*^ mice [5]. Moreover, in *Ig*-*ganp*^Tg^*lyn^−^*^/*−*^ mice, the frequency of cIgM^+^CD11b^+^ cells in the spleen was almost normalized (5.1% in *lyn**^−^*^/*−*^ mice vs. 2.1% in *Ig-ganp*^Tg^*lyn**^−^*^/*−*^ mice; Figure 2D). Thus, biphenotypic cIgM^+^CD11b^+^ cells were mostly observed in *lyn^−^*^/*−*^ mice but not in control or *Ig*-*ganp*^Tg^*lyn^−^*^/*−*^ mice. These results suggest that GANP regulates cell transdifferentiation between B cells and macrophages in a Lyn-independent manner.

### 2.3. Development of B-Cell/Macrophage Biphenotypic Hodgkinoid Lymphoma in Ig-ganp^Tg^ Mice

Long-term observation revealed that lymphoid neoplasms developed only in *Ig-ganp*^Tg^ mice. These neoplasms were frequently observed in the liver and spleen; in addition, neoplastic infiltration of the kidneys and lungs was observed in some cases (Table 1). The histology was classified as malignant lymphoma type and plasma cell myeloma-like type. As reported previously [10], malignant lymphomas developed in *Ig-ganp*^Tg^ mice showed *µ-heavy chain* and *κ-light chain* rearrangements in genomic DNA, expressed µ-/κ-chains, and were immunocytochemically positive for B220 (expressed by the B-cell lineage), only in their cytoplasm. On immunocytochemical examination, we found positive expressions of macrophage-specific markers such as major histocompatibility complex (MHC) class II, F4/80, CD68, and CD204 as well as variable expression levels of cytoplasmic B220 in lymphoid cells (Table 2; Figure 3A,B). These findings indicate that these cells were B-cell/macrophage biphenotypic cells. Reverse transcription-polymerase chain reaction (RT-PCR) revealed negative expressions of *pax5*, *mb-1*, and *Il7R* in the representative *Ig-ganp*^Tg^ mice lymphoma cell line B/M-2; however, we found positive expression of *CD30* (Figure 3C) and strongly positive expression of *c-fms*, similar to that found in the control macrophage cell line J774.1. Fluorescence-activated cell sorting (FACS) also revealed positive expressions of F4/80 and CD204 on the surface of B/M-2 cells (Figure 3D). These results clearly indicate that malignant lymphomas obtained from *Ig-ganp*^Tg^ mice have biphenotypic characteristics of B cells/macrophages and mimic human HL.

### 2.4. Characterization of the B-Cell/Macrophage Biphenotypic Tumor

Subsequently, we performed functional characterization of B-cell/macrophage biphenotypic Hodgkinoid lymphoma using B/M-2. Giemsa staining revealed that B/M-2 cells were larger in size than normal B cells, irregularly shaped, and rich in cytoplasmic vacuoles (Figure 4B, upper panel). Electron microscopic analysis showed many fine cell processes on the surface, which were morphologically akin to those observed in hairy cell leukemia (Figure 4A, lower panel; arrows). Phagocytosis assay using fluorescent microspheres revealed high phagocytic activity of B/M-2 cells (Figure 4B).

Finally, we analyzed the intracellular expressions of cytokines and chemokines by protein array analysis (Figure 4C). Regarding the macrophage profile, B/M-2 cells secreted both cytokines (granulocyte/macrophage colony-stimulating factor (GM-CSF), interleukin (IL)-4, IL-10, IL-12, IL-13, tumor necrosis factor (TNF)-α, and thrombopoietin] and CC chemokines [monocyte chemoattractant protein (MCP)-1, MCP-5, and RANTES). Moreover, we measured the secretion of cytokines and chemokines. Remarkably, MCP-1, macrophage colony-stimulating factor (M-CSF), keratinocyte chemoattractant (KC), RANTES, and vascular endothelial growth factor (VEGF) were secreted in the culture supernatant of B/M-2 cells (Figure 4D). Collectively, these findings suggest that B/M-2 cells exhibit the functionality of macrophages.

## 3. Discussion

In this study, we demonstrated GANP overexpression in human HRS cells and showed that Hodgkinoid lymphomas (characterized by B-cell/macrophage biphenotypic cells) can be developed in *Ig-ganp*^Tg^ mice. B cells and macrophages can be principally distinguished based on the expressions of specific markers: CD45R (B220), CD19, and Ig proteins are B-cell lineage markers, whereas CD11b, CD13, CD15, F4/80, CD68, and CD204 are macrophage lineage markers. B-cell lineage differentiation is committed with the expression of the transcripts of *Pax5*, *mb-1*, *IL-7R*, *RAG1*, and *RAG2*; this proceeds to *Ig*-gene rearrangements, leading to the production of IgM and IgD. Moreover, macrophages are ultimately distinguished by their phagocytic activity and secretion of various inflammatory cytokines [11]. Because the lymphoma cells derived from *Ig-ganp*^Tg^ mice exhibited both *Ig*-gene rearrangements and phagocytic activity, these were characterized as B-cell/macrophage biphenotypic cells with close resemblance to human HL.

We found that B-cell/macrophage biphenotypic Hodgkinoid lymphoma developed in *Ig-ganp*^Tg^ mice had a genetic fingerprint evidence of *Ig*-gene rearrangement with expressions of Ig-µ/Ig-κ chains [12,13,14]. Previous reports suggested the presence of B-cell/macrophage biphenotypic progenitor cells in the fetal liver and less frequently in adult bone marrow [15]; in addition, some researchers believe that HL originates due to the transformation of B-cell/macrophage biphenotypic progenitor cells into bone marrow. Alternatively, we speculate that mature B cells partially transdifferentiate into macrophages in the adult spleen [16], as is also suspected in human HL.

Our study suggests that GANP plays an important role in the transdifferentiation or reprograming of B cells to macrophage-like cells and the consequent development of B-cell/macrophage biphenotypic Hodgkinoid lymphoma that corresponds to human HL. Our previous report indicated that GANP, one of the targets of Lyn-mediated signaling, is most likely regulated via the PU.1 binding site of the *ganp* promoter region [9,17]. Because PU.1 exerts shared transcriptional regulation of both B-cell and macrophage differentiation [18,19], PU.1 may modulate the dynamic reprogramming between B-cell and macrophage differentiation. Indeed, a low concentration of PU.1 leads the fate of B-cell/macrophage biphenotypic precursor cells to B cells, whereas a higher concentration promotes macrophage differentiation and prevents B-cell differentiation [20]. In addition, it is estimated that the amount of *PU.1* mRNA in macrophages is approximately eight times greater than that in B cells [20,21]. Altered signaling through the Lyn-mediated pathway to PU.1-binding sites of the promoter regions in various regulatory molecules may not cause a drastic change in fetal and adult hematopoietic precursor cell differentiation in the liver and bone marrow; however, it may alter germinal center B-cell differentiation in the peripheral lymphoid organs in the humoral immune-deficient state.

Recently, it has gradually been revealed that GANP possesses multiple functions. Previous report indicated that GANP upregulation is essential for the survival of mature germinal center B-cells with high affinity type due to suppression of DNA damages [9]. Taken together with the previous and present results, GANP may also be required for the survival of HRS cells originated from germinal center B-cells of *lg-ganp*^Tg^ mice. Hence, we speculate that both survival and maintenance of germinal center B-cells and transdifferentiation between B-cells and macrophages by GANP are synergistically related to Hodgkin lymphomagenesis.

We could not demonstrate transplantability of these Hodgkinoid lymphoma cells in tumor xenograft experiments on C57BL/6 and immunodeficient mice, although we challenged different inoculation methods such as multiple subcutaneous and intrasplenic injections. Previous reports indicate that HRS cells grow in a typical microenvironment composed of many different types of leukocytes such as B cells, T cells, and eosinophils, which is most likely essential for HRS cell survival [22]. HRS cells are considered to regulate their microenvironment and attract many infiltrating cells specifically by secreting cytokines and chemokines [22]. Accordingly, *lg*-*ganp*^Tg^ mice-derived Hodgkinoid lymphoma cells proliferate slowly even in the presence of M-CSF. Thus, transplantability to C57BL/6 or immunodeficient mice seems to be relatively difficult because Hodgkinoid lymphoma cells may have features similar to HRS cells.

Notably, 40% of human classical HL is associated with EBV infection. Because EBV can immortalize B cells in vitro, it plays an important role in Hodgkin lymphomagenesis. Latent membrane protein 1, one of the latent EBV genes, mimics an active CD40 receptor [23]. CD40 expressed on HRS cells activates NF-κB signaling constitutively, and CD40 signaling may be critical for Hodgkin lymphomagenesis in *Ig*-*ganp*^Tg^ mice because our murine Hodgkinoid lymphoma is not infected by EBV. Consistent with this notion, GANP is known to operate downstream of CD40. Thus, the upstream or downstream molecules of GANP may be alternative therapeutic targets of HL and CD30.

Overall, we established a cytological model of HL while focusing on its biphenotypic characteristics of B cells/macrophages and GANP, a signaling molecule that may modulate dynamic transdifferentiation and reprogramming of B cells to macrophages. To the best of our knowledge, this is the first cytological model of HL, although some non-HL models have already been established [24]. *Ig*-*ganp*^Tg^ mice may serve as a novel cytological model that may facilitate the study of cytopathological etiology and oncogenesis of HL.

## 4. Materials and Methods

### 4.1. Mice

*lyn*^+/*−*^, *lyn^−^*^/*−*^, and *Ig*-*ganp*^Tg^ mice with C57BL/6 background were maintained in specific pathogen-free conditions at the Center for Animal Resources and Development, Kumamoto University. The procedure of animal experiments was approved by the ethics committee of Kumamoto University (approval number: A25-091) based on the Fundamental Guidelines of Conduct of Animal Experiment and Related Activities in Academic Research Institutions under the jurisdiction of the Ministry of Education, Culture, Sports, Science and Technology in Japan.

### 4.2. Cells and Cell Culture

B/M-2 was established from the lymphoma lesion of *Ig*-*ganp*^Tg^ mice [10]. The mouse pre-B-cell line 70Z/3 and macrophage cell line J774.1 were used as controls. The human HL cell lines L428 and HD70 were purchased from the JCBR Cell Bank (National Institutes of Biomedical Innovation, Health, and Nutrition, Osaka, Japan). All cell lines were cultured in RPMI-1640 medium containing 10% fetal calf serum (Sigma-Aldrich, St. Louis, MO, USA). Cell morphology was analyzed by Wright–Giemsa staining (Muto Chemicals, Tokyo, Japan).

### 4.3. Preparation of Cell Blocks

L428 or HD70 was harvested, washed with phosphate-buffered saline (PBS), and resuspended in 10% neutral buffered formalin (Wako Pure Chemicals, Osaka, Japan) for fixation. After the supernatant was discarded by centrifugation, 1% sodium alginate (Wako) and then 1% calcium chloride solution (Wako) was added to the sediment. The gelated sample was treated along with other routine histopathological specimens. The formalin-fixed paraffin-embedded cell blocks were sectioned into 4-μm sections and used for further analyses.

### 4.4. Immunocytochemical Analysis

Regarding the cell blocks of L428 and HD70, immunocytochemistry with an anti-GANP monoclonal antibody (42-23 [25]) was performed using the labeled as the streptavidin–biotin (LSAB) staining method, as described previously [26,27]. A negative control experiment was carried out by omitting an anti-GANP monoclonal antibody from the LSAB staining method. Conventional immunocytochemistry for CD15 (MMA, mouse IgM; Roche Diagnostics, Basel, Switzerland) and CD30 (Ber-H2, mouse IgG; Roche Diagnostics) was performed using the iView DAB Universal Kits run on Ventana Benchmark Ultra (Roche Diagnostics).

Regarding tumors in *Ig*-*ganp*^Tg^ mice, conventional immunocytochemistry was performed using the indirect method. Formalin-fixed, paraffin-embedded sections were deparaffinized, blocked with Blocking-One (Nacalai Tesque, Kyoto, Japan), and incubated with primary antibodies against B220 (RA3/6B2 [26]), F4/80 (A3-1, rat IgG2b; Abcam, Cambridge, UK), CD68 (FA-11, rat IgG2; Abcam), GL7 (GL7, rat IgMκ; BD Biosciences, Mountain View, CA, USA), CD3 (2C11 [25]), µ (AM/3 [25]), κ (HB58 [25]), CD5 (53-7.3, rat IgG2aκ; BD Biosciences), CD138 (EPR6454, rabbit IgG; Abcam), bcl-6 (rabbit polyclonal; Santa Cruz Biotechnology, Santa Cruz, CA, USA), and MHC class II (NIMR-4, rat IgG2b; Abcam) in combination with an appropriate horseradish peroxidase (HRP)-conjugated anti-rat IgG antibody (Southern Biotechnology Associates, Birmingham, AL, USA), anti-rat IgM antibody (Invitrogen, Carlsbad, CA, USA), or anti-rabbit IgG antibody (Invitrogen).

Regarding B/M-2 culture cells, cytological immunofluorescence was performed. Cells were cultured in Lab-Tek chamber slides (Nalge Nunc, Penfield, NY, USA), fixed with 4% paraformaldehyde/PBS, and permeablized with 0.2% Triton X-100. After blocking, cell smears were incubated with an anti-F4/80 monoclonal antibody in combination with an Alexa488-conjugated anti-rat IgG antibody. Subsequently, biotin-labeled anti-B220 or GANP monoclonal antibody [25] in combination with Alexa 546-conjugated streptavidin (Life Technologies, Carlsbad, CA, USA) was used. Signals were captured using the confocal laser microscope FV500 (Olympus, Tokyo, Japan).

### 4.5. Flow Cytometry

Lymphoid cells from the bone marrow and spleen and B/M-2 cells were stained with an fluorescein isothiocyanate-conjugated monoclonal antibody, phycoerythrin-conjugated monoclonal antibody, and allophycocyanin-conjugated monoclonal antibody as well as biotin-labeled monoclonal antibody with PerCP-conjugated streptavidin (GE Healthcare, Buckinghamshire, UK). Fluorescent dye-conjugated monoclonal antibodies were purchased from eBioscience (San Diego, CA, USA; for anti-B220, IgM, IgD, and CD11b antibodies), BD Biosciences (for anti-CD43 antibody), and AbD Serotec (Kidlington, UK; for anti-F4/80 and CD204 antibodies). Biotin-labeled monoclonal antibodies were produced using hybridoma cells (for anti-B220 and µ antibodies [25]). For staining cIgM, cells were fixed and permeabilized with BD Cytofix/Cytoperm (BD Biosciences). Data were analyzed by FACSCalibur (BD Biosciences, Mountain View, CA, USA) using FlowJo (Tree Star, Ashland, OR, USA).

### 4.6. RT-PCR

cDNA synthesis and PCR were conducted as described previously [10]. Primers used in this study are listed in Table 3.

### 4.7. Electron Microscopic Analysis

B/M-2 cells were fixed with 2.5% glutaraldehyde in 100 mM cacodylate buffer and post-fixed with 1% osmium tetroxide. After dehydration by passage through graded ethanol series, samples were embedded in epoxy resin. Ultrathin sections were stained with uranyl acetate and lead citrate and examined under the H-7500 electron microscope (Hitachi, Tokyo, Japan).

### 4.8. Phagocytosis Assay

B/M-2 cells were cultured on a six-well tissue culture plate and incubated with fluorescent microspheres (Fluoresbrite Carboxylate Microspheres; diameter: 0.7 µm; Polysciences, Warrington, PA) for 5 h at 37 °C with 5% CO_2_. After several washes with PBS to remove non-phagocytosed beads, cells were harvested using Cell Dissociation Buffer (Life Technologies). Cells showing phagocytized particles were analyzed by FACSCalibur.

### 4.9. Cytokine Production

B/M-2 cells were washed with PBS and then incubated with Cell Lysis Buffer (Mouse Cytokine Array kit; RayBiotech, Norcross, GA, USA). After centrifugation, cell lysate was recovered from the supernatant. The array membrane was incubated with cell lysate and biotin-conjugated detection antibodies in combination with HRP-conjugated streptavidin. Development was performed using an enhanced chemiluminescence detection system (GE Healthcare).

To measure the concentrations of cytokines and chemokines in the culture supernatant, B/M-2 cells were cultured overnight in serum-free media (complete DPM; Life Technologies) at 37 °C. The supernatant was centrifuged at 10,000× *g* for 15 min at 4 °C. The concentrations of various cytokines and chemokines were measured using the Bio-Plex Pro assay (Bio-Rad, Hercules, CA, USA).

## 5. Conclusions

Cytological molecular analysis of HL is challenging because there is no established murine model. B-cell/macrophage biphenotypic cells were found in the spleen of *lyn*^−/−^ mice. HL cells overexpress Lyn-target GANP and exhibit unique biphenotypic characteristics of B cells/macrophages. Moreover, *Ig-ganp*^Tg^ mice have been found to spontaneously develop Hodgkinoid tumors that exhibit biphenotypic characteristics of B cells/macrophages corresponding to human HL. More interestingly, GANP overexpression might regulate transdifferentiation between B cells and macrophages. To the best of our knowledge, this is the first cytological model of HL. *Ig*-*ganp*^Tg^ mice might help in the study of the cytopathological etiology and lymphomagenesis of HL.

## Figures and Tables

**Figure 1 cancers-12-00204-f001:**
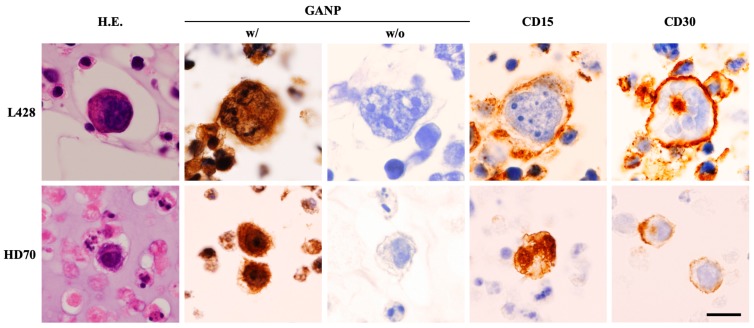
Hematoxylin and eosin staining and immunocytochemistry of the Hodgkin lymphoma cell lines L428 and HD70. Both cell lines have medium-to-large-sized cells exhibiting a bizarre nucleus with a large nucleolus. Multinucleated Reed–Sternberg cells are observed. Immunocytochemically, these cells are positive not only for the diagnostic markers CD15 and CD30 but also for germinal center-associated nuclear protein (GANP). Immunoreactivity is completely lost when an anti-GANP monoclonal antibody (42-23) is omitted from the immunocytochemical procedure (negative control). Scale bar = 25 µm.

**Figure 2 cancers-12-00204-f002:**
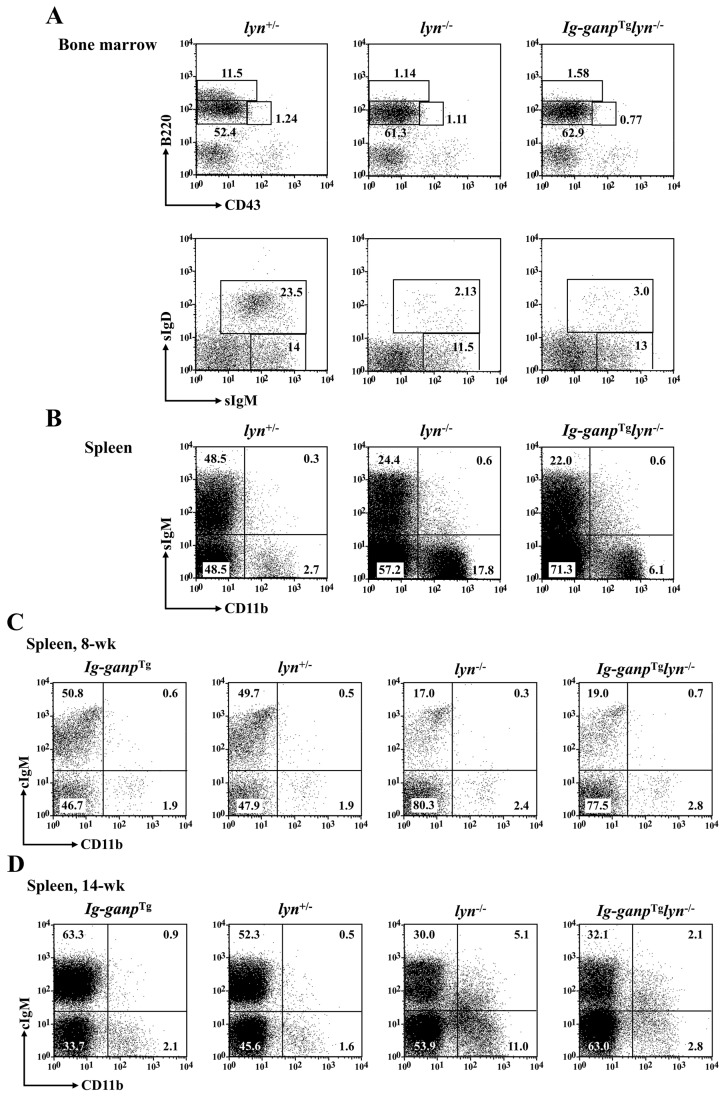
Cell differentiation into B-cell/macrophage biphenotypic cells by GANP in a Lyn-deficient state. Early B-cell differentiation compared among *lyn*^+/*−*^, *lyn^−^*^/*−*^, and *Ig*-*ganp*^Tg^*lyn^−^*^/*−*^ mice. (**A**) Bone marrow cells isolated from 14-week-old mice were stained with B220, CD43, IgM, and IgD to identify pro-B, pre-B, immature B, and mature B-cell fractions. Although there are no differences of pro-B, pre-B, and immature B-cell populations, sIgM^+^sIgD^+^ mature B-cell population is reduced in *lyn**^−^*^/*−*^ and *Ig-ganp*^Tg^*lyn**^−^*^/*−*^ mice. (**B**) sIgM^−^CD11b^+^ population in the spleen is reduced in *Ig*-*ganp*^Tg^*lyn^−^*^/*−*^ mice compared to *lyn^−^*^/*−*^ mice. (**C**,**D**) cIgM^+^CD11b^+^ cell population is increased in 14-week-old *lyn**^−^/^−^* mice, whereas the population is almost normal in 14-week-old *Ig-ganp*^Tg^*lyn**^−^*^/*−*^ mice.

**Figure 3 cancers-12-00204-f003:**
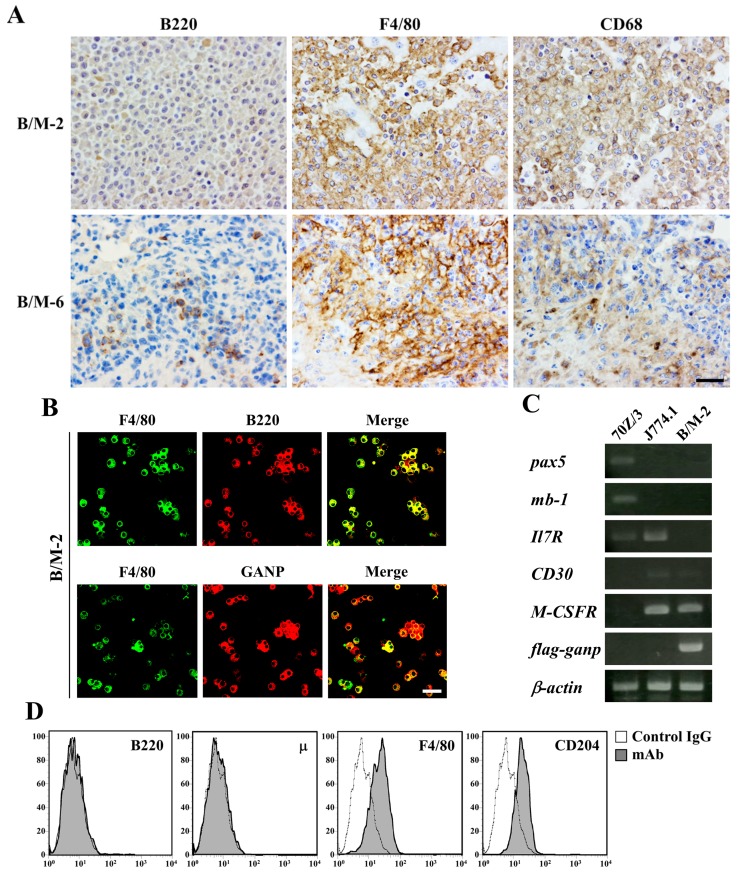
B-cell lineage and macrophage markers in lymphomas in *Ig*-*ganp*^Tg^ mice. (**A**) Immunocytochemical staining of B/M-2 and B/M-6 cells. Both B-cell lineage marker B220 and macrophage markers F4/80 and CD68 are copositive. Scale bar = 20 µm. (**B**) Cytological immunofluorescence on B/M2 cell line. The macrophage marker F4/80, B-cell lineage marker B220, and GANP are colocalized. Scale bar = 20 µm. (**C**) Expression of various transcripts related to B cells (*pax5*, *mb-1*, and *Il7R*) and macrophages (*CD30* and *c-fms*) in B/M-2. 70Z/3 and J774.1 are used as controls for B cells and macrophages, respectively. Exogenous *ganp* transcripts are detected using *flag*- and *ganp*-specific polymerase chain reaction primers. *β-Actin* was used as a loading control. (**D**) Surface expression of various markers on B/M-2. These data collectively suggest that B220 is expressed not on the surface but in the cytoplasm. All data are representative of three independent experiments.

**Figure 4 cancers-12-00204-f004:**
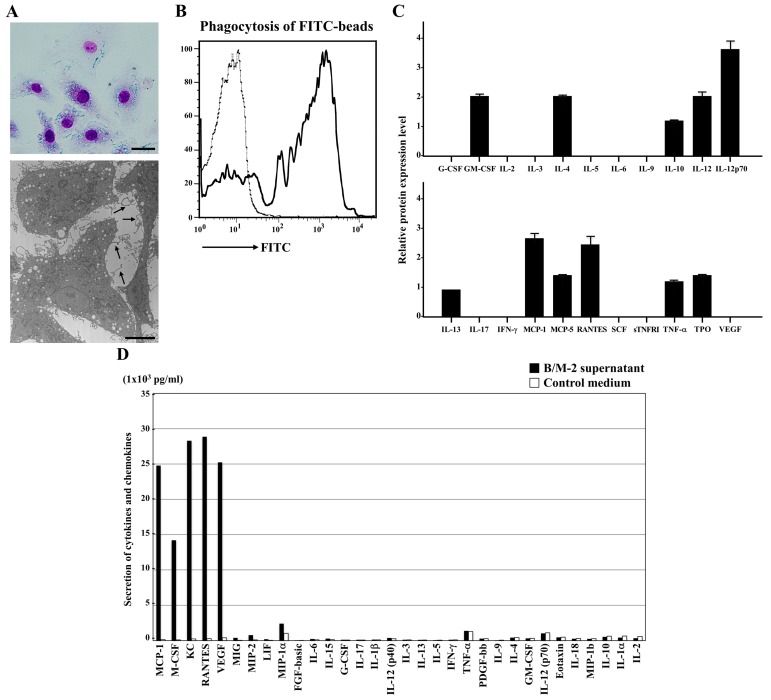
Morphological and functional properties of the Hodgkinoid cell line B/M-2 derived from *Ig*-*ganp*^Tg^ mice. (**A**) A representative B/M-2 is morphologically analyzed by Wright–Giemsa staining (upper panel; scale bar = 20 µm) and electron microscopy (lower panel; scale bar = 7 µm). B/M-2 cells are larger in size, irregularly shaped, rich in vacuoles in the cytoplasm, and exhibit many fine cell processes on the surface (arrows). (**B**) The phagocytic activity in B/M-2 cells was determined by measuring the uptake of fluorescein isothiocyanate (FITC)–microspheres. Cells cultured in the absence of FITC–microspheres are used as negative control (thin line). (**C**) Measurement of cytokine expression in B/M-2 cells. Signal intensity is calculated using a densitometer, and the value of IL-13 is defined as 1.0 for relative comparison. (**D**) Cytokine/chemokine secretion from B/M-2 cells. A serum-free medium is used as a control.

**Table 1 cancers-12-00204-t001:** Organs of tumor development in *Ig*-*ganp*^Tg^ mice.

Tumor ID	Spleen	Liver	LNs	Pathological Features
B/M-1	++	+	++	Malignant lymphoma (DLBCL)
B/M-2	+	++	++	Malignant lymphoma (B lineage) Liver: plasmacytoma (myeloma like)
B/M-3	−	++	+/−	Malignant lymphoma (B lineage) Kidneys, lungs: lymphoma invasion +
B/M-4	++	+	++	Malignant lymphoma (DLBCL) Kidneys, lungs: lymphoma invasion +
B/M-5	−	+	n.d.	Malignant lymphoma (lineage undetermined) Liver: lymphoma invasion
B/M-6	+/−	++	n.d.	Malignant lymphoma (B lineage)
B/M-7	−	++	n.d.	Malignant lymphoma (B lineage) Liver: plasmacytoma (myeloma like)
B/M-9	−	++	n.d.	Malignant lymphoma (B lineage) Liver: plasmacytoma (myeloma like)
B/M-10	−	+	n.d.	Malignant lymphoma (lineage undetermined)

Note: Tumor IDs were named arbitrarily. Involvement of each organ by tumor was evaluated as follows: −, none; +/*−*, <10%; +, 10–40%; ++, >40%. Abbreviations: LNs, lymph nodes; DLBCL, diffuse large B-cell lymphoma; n.d., not determined.

**Table 2 cancers-12-00204-t002:** Immunocytochemical analysis of Hodgkinoid lymphoma cell lines derived from *Ig*-*ganp*^Tg^ mice.

Tumor ID	Surface Phenotype by FACS	µ	κ	B220	CD5	CD138	Bcl-6	MHC Class II	F4/80	CD68	CD204
B/M-1	B220^+^IgM^+^GL7^+^	+/−	+/−	+	−	−	+	++	++	++	++
B/M-2	B220^−^IgM^−^CD3*^−^*	+/−	+/−	+	−	−	+/−	++	+	+	+
B/M-3	B220^−^IgM^−^CD3*^−^*	+/−	+/−	+	−	−	+/−	++	++	++	++
B/M-4	B220^−^IgM^−^CD3*^−^*	+/−	+/−	+	−	−	n.d.	++	++	++	++
B/M-6	B220^−^IgM^−^CD3*^−^*	+/−	+/−	+/−	−	−	n.d.	++	+	+	+

Note: Tumor IDs were named arbitrarily. +/−, weakly positive; +, moderately positive; ++, strongly positive. Abbreviations: FACS, fluorescence-activated cell sorting; MHC, major histocompatibility complex; n.d., not determined.

**Table 3 cancers-12-00204-t003:** Primer sequences.

Primers	Orientation	Sequence (5′-3′)
Mouse *pax5*	Forward	CTACAGGCTCCGTGACGCAG
	Reverse	GTCTCGGCCTGTGACAATAGG
Mouse *mb-1*	Forward	GCCAGGGGGTCTAGAAGC
	Reverse	TCACTTGGCACCCAGTACAA
Mouse *Il7R*	Forward	CGAGTGAAATGCCTAACTC
	Reverse	GCGTCCAGTTGCTTTCAC
Mouse *CD30*	Forward	GATTCCTGTCCTACTGAAAAGCTA
	Reverse	TTGTCACTTCTCAGAGACAGTCGT
Mouse *c-fms*	Forward	TCATTCAGAGCCAGCTGCCCAT
	Reverse	ACAGGCTCCCAAGAGGTTGACT
Mouse *flag-tagged ganp*	Forward	GATTACAAGGATGACGACGATAAG
	Reverse	GCGCACAGACTTTCCCCTGA

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
