# Peer review of "A Novel Cytological Model of B-Cell/Macrophage Biphenotypic Cell Hodgkin Lymphoma in Ganp-Transgenic Mice"

_cancers, 2020, doi:10.3390/cancers12010204_

Round 1

Reviewer 1 Report

In this manuscript, Sakai et al. demonstrate that germinal center-associated nuclear protein (GANP) is overexpressed in human Hodgkin lymphoma (HL) cells and that Ganp transgenic mice (lg-ganpTg mice) develop lymphomas with B/macrophage biphenotypic characteristics just like HL cells. Data presented in the manuscript are well documented and establishing lg-ganpTg mice as a cytologic model of HL is certainly a meaningful progress of the field. However, there are a couple of concerns that need to be addressed by the authors.

The authors state that GANP may regulate transdifferentiation between B-cells and macrophages. Is GANP overexpression required for maintaining the biphenotypic characteristics in HL cells? What would be the consequence of knocking down GANP in human HL cells or in lymphoma cell lines derived from lg-ganpTg mice? Immunohistochemistry in Figure 1 requires negative-control cells such as other types of lymphoma cells that do not overexpress GANP. To establish lg-ganpTg mice as a cytologic model of HL, it would be essential to demonstrate the transplantability of the lg-ganpTg-derived Hodgkinoid lymphoma cells in B6 mice or in immunocompromised mice.

Author Response

Response to Reviewer 1 Comments

In this manuscript, Sakai et al. demonstrate that germinal center-associated nuclear protein (GANP) is overexpressed in human Hodgkin lymphoma (HL) cells and that Ganp transgenic mice (lg-ganpTg mice) develop lymphomas with B/macrophage biphenotypic characteristics just like HL cells. Data presented in the manuscript are well documented and establishing lg-ganpTg mice as a cytologic model of HL is certainly a meaningful progress of the field. However, there are a couple of concerns that need to be addressed by the authors.

We thank the referees for providing the important criticisms concerning our manuscript. We tried to answer all the questions, criticisms, and suggestions by changing the descriptions. We believe that all the comments and questions were answered accordingly. We confirmed that all authors have approved this change in the revised manuscript.

Point 1: The authors state that GANP may regulate transdifferentiation between B-cells and macrophages. Is GANP overexpression required for maintaining the biphenotypic characteristics in HL cells?

Response 1: Thank you for your critical comments. Recently, it has gradually been revealed that GANP possesses multiple functions. Previous report indicated that GANP upregulation is essential for the survival of mature germinal center B-cells with high affinity type due to the suppression of DNA damages [Kuwahara et al., J. Immunol., 2012]. Taken together with the previous and present results, GANP may be also required for the survival of HRS cells originated from germinal center B-cells of lg-ganpTg mice. Hence, we speculate that both the survival and maintenance of germinal center B-cells and transdifferentiation between B-cells and macrophages by GANP are synergistically related to Hodgkin lymphomagenesis. These descriptions are added to the Discussion section (page 8, lines 229 to 236).

Point 2: What would be the consequence of knocking down GANP in human HL cells or in lymphoma cell lines derived from lg-ganpTg mice?

Response 2: Unfortunately, we could not knockdown GANP in human HL cells (L428 and HD70) and lymphoma cell lines derived from lg-ganpTg mice, because of very low transfection efficiency even using RNAiMAX (Life Technologies). Especially, the latter cell lines are difficult to knockdown ganp gene, because of growth retardation. In our previous studies, human B cell lines such as Daudi and Ramos after ganp depletion induced cell-cycle retardation leading to apoptosis. This phenomenon is observed in many types of cell lines through RNA integrity [Yoshida et al., Genes Cells, 2007; Okamoto et al., Genes Cells, 2010]. Based on this information, HL cells after ganp depletion would show cell-cycle arrest leading to apoptosis.

Point 3: Immunohistochemistry in Figure 1 requires negative-control cells such as other types of lymphoma cells that do not overexpress GANP.

Response 3: Thank you for your important comments. As shown in ref. 23, ganp transcripts are ubiquitously detected in various tissues. In addition, GANP protein is aberrantly expressed in many types of tumors [ref. 10; Kageshita et al., J. Dermatol. Sci., 2006; Ohta et al., Cancer Sci., 2009; Chan-on et al., Int. J. Oncol., 2009]. We have examined GANP expression using several lymphoma cell lines and confirmed that all cell lines we checked were GANP-positive to a lesser degree. Therefore, it is difficult to prepare negative-control lymphoma cells lacking GANP expression as the reviewer indicated. Instead, we added negative-control experiments without anti-GANP mAb to show that GANP was definitely expressed in HL cell lines (Figure 1). In this figure, we do not mean to argue the specific expression of GANP in HL cells. These descriptions are added to the legend for Figure 1 (page 3, lines 100 to 101) and Materials and Methods section (page 9, line 285 to 287).

Point 4: To establish lg-ganpTg mice as a cytologic model of HL, it would be essential to demonstrate the transplantability of the lg-ganpTg-derived Hodgkinoid lymphoma cells in B6 mice or in immunocompromised mice. 

Response 4: Thank you for your critical concerns. However, we could not demonstrate transplantability of these Hodgkinoid lymphoma cells in tumor xenograft experiments on C57BL/6 and immunodeficient mice, although we challenged the different inoculation methods such as subcutaneous and intrasplenic injection several times. Previous reports indicated that HRS cells grow into a typical microenvironment composed of many different types of leukocytes such as B cells, T cells and eosinophils, which is most likely essential for HRS cell survival [Küppers et al., Nat. Rev. Cancer, 2009]. HRS cells are considered to regulate their microenvironment and attract many of the infiltrating cells specifically by the secretion of cytokines and chemokines. Accordingly, lg-ganpTg mice-derived Hodgkinoid lymphoma cells proliferate slowly even in the presence of M-CSF. Thus, transplantability to C57BL/6 or immunodeficient mice seems to be quite difficult since the Hodgkinoid lymphoma cells may have similar features with HRS cells. These descriptions are added to the Discussion section (page 8, lines 237 to 246).

Reviewer 2 Report

The authors tested succcessfully a new model of of B-cell/macrophage 2 biphenotypic Hodgkin Lymphoma in Ganp 3 gene-transgenic mice.

In the methodological part, they should mention what was the cell block methodology.

The discussion part could contain practical issues of cytopathological molecular reserach and Hodgkin lymphoma.

Author Response

Response to Reviewer 2 Comments

Point 1: In the methodological part, they should mention what was the cell block methodology.

Response 1: In response to the comment, we described the cell block technique in Materials and Methods (page 9, line 275 to 281).

Point 2: The discussion part could contain practical issues of cytopathological molecular research and Hodgkin lymphoma.

Response 2: Thank you for your critical comments. We added two paragraphs concerning practical issues of cytopathological molecular research and Hodgkin lymphoma (pages 8 to 9, lines 237 to 254).

Round 2

Reviewer 1 Report

The authors adequately, if not completely, addressed my concerns. The manuscript is now suited for publication in Cancers.